# Isatin Bis-Imidathiazole Hybrids Identified as FtsZ Inhibitors with On-Target Activity Against *Staphylococcus aureus*

**DOI:** 10.3390/antibiotics13100992

**Published:** 2024-10-19

**Authors:** Rita Morigi, Daniele Esposito, Matteo Calvaresi, Tainah Dorina Marforio, Giovanna Angela Gentilomi, Francesca Bonvicini, Alessandra Locatelli

**Affiliations:** 1Department of Pharmacy and Biotechnology, Alma Mater Studiorum-University of Bologna, Via Belmeloro 6, 40126 Bologna, Italy; rita.morigi@unibo.it (R.M.); daniele.esposito6@unibo.it (D.E.); alessandra.locatelli@unibo.it (A.L.); 2Department of Chemistry “Giacomo Ciamician”, Alma Mater Studiorum-University of Bologna, Via Selmi 2, 40126 Bologna, Italy; matteo.calvaresi3@unibo.it (M.C.); tainah.marforio2@unibo.it (T.D.M.); 3IRCCS Azienda Ospedaliero-Universitaria di Bologna, Via Massarenti 9, 40138 Bologna, Italy; 4Department of Pharmacy and Biotechnology, Alma Mater Studiorum-University of Bologna, Via Massarenti 9, 40138 Bologna, Italy; giovanna.gentilomi@unibo.it; 5Division of Microbiology, IRCCS Azienda Ospedaliero-Universitaria di Bologna, Via Massarenti 9, 40138 Bologna, Italy

**Keywords:** isatin bis-imidathiazole hybrids, antibacterial activity, MRSA, FtsZ inhibitors, molecular dynamics simulations

## Abstract

In the present study, a series of isatin bis-imidathiazole hybrids was designed and synthesized to develop a new class of heterocyclic compounds with improved antimicrobial activity against pathogens responsible for hospital- and community-acquired infections. A remarkable inhibitory activity against *Staphylococcus aureus* was demonstrated for a subset of compounds (range: 13.8–90.1 µM) in the absence of toxicity towards epithelial cells and human red blood cells. The best performing derivative was further investigated to measure its anti-biofilm potential and its effectiveness against methicillin-resistant *Staphylococcus aureus* strains. A structure–activity relationship study of the synthesized molecules led to the recognition of some important structural requirements for the observed antibacterial activity. Molecular docking followed by molecular dynamics (MD) simulations identified the binding site of the active compound FtsZ, a key protein in bacterial cell division, and the mechanism of action, i.e., the inhibition of its polymerization. The overall results may pave the way for a further rational development of isatin hybrids as FtsZ inhibitors, with a broader spectrum of activity against human pathogens and higher potency.

## 1. Introduction

The emergence and spread of resistant bacteria, coupled with the paucity of new antibiotics, has evolved into a global health crisis and, as a result, infectious diseases have become more challenging or even impossible to treat, leading to an increase in morbidity and mortality [1]. To combat multi-drug-resistant infectious microorganisms, many scientists worldwide are focusing on the development of effective antimicrobial drugs, alternative to traditional antibiotics. Historically, all known clinical antibiotics target bacterial structures and cellular processes such as cell membranes, cell walls, DNA or RNA or protein synthesis, as well as metabolic compound synthesis [2,3]. Unfortunately, many bacterial strains have currently become resistant to all these existing antibiotics [4].

In search of new classes of antimicrobials, with a novel mode of actions, bacterial cytoskeleton and biochemistry events related to cell division are considered as prime targets [3,5]; in particular, the cytoskeletal protein FtsZ, filamentous temperature-sensitive mutant Z, has gained much recognition in the development of antimicrobial therapeutics [6,7]. Indeed, FtsZ is an ideal drug target due to the highly conserved protein found in bacteria despite the structural and functional similarities. It is a distant ancestral homologue of the eukaryotic β-tubulin with <20% sequence identity, thus reducing the likelihood that molecules targeting FtsZ may affect mammalian cells.

FtsZ is an essential bacterial GTPase that plays a leading role in cell division machinery; at the earliest step of the process, FtsZ undergoes assembly at mid-cell, forms a filamentous ring-like structure called Z-ring via GTP-dependent polymerization, and, following the recruitment of other necessary bacterial division proteins, the dynamic Z-ring constricts to generate the two daughter cells. Bacterial cells depleted of or lacking functional FtsZ do not divide and eventually lyse [5].

Among the FtsZ inhibitors recently reported [8,9,10], heterocyclic compounds such as isatin (indoline-2,3-dione) and its derivatives have exhibited antibacterial activities [11]. As a “privileged building block”, almost all positions in the isatin moiety can be modified resulting in several molecules with a broad spectrum of biological properties including antitumor drugs, anticonvulsants, anti-inflammatory and antioxidant agents, and antimicrobials [12,13,14].

Considering the versatility of the isatin precursor in pharmacophore modeling and the promising results previously obtained [15] that led to identification of isatin derivatives active against *Staphylococcus aureus*, in the present study, the chemical space for isatin bis-imidazothiazole hybrids was further explored by planning the development of a new series of 18 derivatives.

First, the analogs of the two lead compounds, **6k** and **6m** ([Fig antibiotics-13-00992-ch001]) [15], were synthetized, introducing a methyl group at position 2 of the 6-chloroimidazo[2,1-b]thiazole portion (compounds **3**–**4**) to compare activity. Subsequently, different substituents were introduced on the isatin moiety to obtain **6k** analogs, such as fluorine at position 5 (compounds **5**–**6**) or 6 (**7**–**8**), iodine and OCF_3_ at position 6 (**9**–**10** and **11**–**12** respectively), while keeping the 6-chloroimidazo[2,1-b]thiazole and its 2-methyl analog in position 3.

Regarding **6m** analogs, substituted on isatin nitrogen, compounds **13**–**14** and **15**–**16** were synthesized bearing the aforementioned imidazothiazole portions to evaluate the effect of introducing benzyl and prop-1-en-1-ylbenzyl groups, instead of 4-chlorobenzyl. Finally, in an attempt to obtain compounds with iodine or CF_3_ at position 4 of the isatin nucleus, only the mono-imidazothiazole hybrids’ derivatives **17**–**18** and **19**–**20** were obtained, characterized, and tested to obtain preliminary information about the absence of one imidazothiazole nucleus.

The present work deals with the design, synthesis, biological evaluation, and molecular dynamics (MD) studies of a new class of isatin bis-imidazothiazole hybrids able to inhibit bacterial growth by targeting FtsZ protein.

## 2. Results and Discussion

### 2.1. Chemistry

The new compounds **3**–**20** were synthesized from appropriately substituted isatin (compounds **2c**–**l**, an equivalent) and the appropriate imidazothiazole (compounds **1a**–**b**, two equivalents) previously synthesized, using molecular iodine as a catalyst in isopropanol [16]. Only for the synthesis of compounds **17** and **19,** the synthetic strategy was changed in order to achieve isatin bis-imidazothiazole hybrids, using iron(III) chloride as a catalyst in acetonitrile [17]. However, only the monoderivatives were obtained in this case too (Figure 1).

The 6-chloroimidazo[2,1-b]thiazoles, **1a**–**l**, were prepared as described in the literature. The isatins, **2c**–**l**, were either commercially available or prepared via the Sandmeyer reaction of aniline derivatives with chloral hydrate and hydroxylamine hydrochloride followed by cyclization in concentrated sulfuric acid.

The structures of the final compounds were confirmed by means of ^1^H-NMR, ^13^C-NMR, and UPLC MS spectra (Appendix A).

The analysis of the ^1^H-NMR spectra allowed us to make the following considerations: the compounds with the isatin core bearing two imidazothiazoles have no plane of symmetry; thus, the imidazothiazoles are in different chemical environments. Indeed, each imidazothiazole proton gives its own signal, and in compounds **13**–**16,** the benzylic methylene protons form two coupling doublets instead of a singlet because they are diastereotopic.

### 2.2. Biological Evaluation

The antimicrobial activity of the newly designed hybrid compounds was evaluated in vitro against three reference strains including Gram-positive and Gram-negative bacteria (*S. aureus* ATCC 25923 and *E. coli* ATCC 25922, respectively) and fungi (*C. albicans* ATCC 10231). The biological studies also included the assessment of the cytotoxicity on epithelial cells by measuring the viability of Vero cell line. All compounds were screened for their effects on these model systems at 100 μM; subsequently, the best-performing compounds were assayed to determine their IC_50_ and CC_50_ values and their hemolytic properties on hRBCs, thus obtaining a complete overview on their overall potential as antimicrobials.

### 2.3. In Vitro Antimicrobial Study

The antimicrobial activity of the isatin-based derivatives was assayed by a standardized microdilution method; the cellular growth measured in the different model systems, when treated with 100 μM of compound, is reported in Table 1.

Some general remarks can be drawn from data reported in Table 1. As for the first series of isatin hybrids previously obtained [15], none of the newly synthetized derivatives displayed an inhibitory activity against *E. coli* and *C. albicans* at 100 μM, but they affected Vero cell metabolism to a variable extent. In detail, considering the percentage values, compounds **5**–**6** and **11**–**13** reduced cellular viability, with compound **11** being the most toxic and **13** the less active in this cellular environment.

As compounds **5**, **11,** and **15** turned out to strongly decrease *S. aureus* growth (<30% compared to the positive control), they warranted further investigation to delve into their antibacterial activity as well as their selectivity. These isatin bis-imidazothiazole hybrids were subjected to dose–response experiments on *S. aureus* to measure their IC_50_ values along with cell viability assays on Vero cells to determine their effect towards mammalian cells. Thus, IC_50_ values were analyzed in light of the corresponding CC_50_ values and reported in Table 2.

While the IC_50_ value of compound **5** (75.6 µM) was very close to the lead **6k** (IC_50_ = 79.95 µM) [15], indicating that the modification made on the core structure did not result in a significant increase in the antibacterial efficacy, and compound **15** showed only a moderate inhibitory activity, the IC_50_ of compound **11** showed a 6-fold decrease with a value of 13.8 µM. Notably, the activity of the compound also maintained a favorable selectivity towards bacterial cells without adverse effects on the tested mammalian cells at the inhibitory concentration. Indeed, the CC_50_ of compound **11** was 63.3 µM with an SI of 4.6.

Considering that the lack of selectivity and the cytotoxicity could represent a limiting factor in the antimicrobial agent discovery pipeline, compound **11** was also investigated for its hemolytic effect on hRBCs. The hemolysis assay is generally intended as method for the evaluation of lytic interactions with mammalian membranes, thus providing information on the overall therapeutic value of the drug [18]. Compound **11**, assayed in the range 100–1.56 µM, did not show an appreciable hemolytic activity against human erythrocytes (Appendix A), thus revealing its general systemic safety, even at the highest concentrations.

As a proof of concept, to evaluate the overall potency of compound **11**, isolates of methicillin-resistant *S. aureus* (MRSA) with a different antibiotic resistance profile were tested and their results compared with the ATCC strain (Appendix A). It is worth noting that the IC_50_ values obtained for these strains were close to those of the reference control as demonstrated in Figure 1. This is clinically relevant considering that isolates may present phenotypic and genetic heterogeneity compared to laboratory reference strains, thus some diversities in susceptibility may occur.

### 2.4. Molecular Dynamics Simulations

Isatin derivatives have already demonstrated antibacterial activity, in particular as FtsZ inhibitors [11]. FtsZ is a key organizing protein in bacterial cell division and the prokaryotic equivalent of the eukaryotic cytoskeletal tubulin. FtsZ polymerizes by a head-to-tail interaction with tubulin-like protofilaments during bacterial cell division. Since FtsZ is necessary for bacterial cell division in nearly all prokaryotes, but it is missing in eukaryotes, it has been identified as a novel target for antibacterial drugs. Anti-FtsZ drugs block FtsZ polymerization, impeding the binding of the critical GTP cofactor or obstructing FtsZ subunit interactions sites (Figure 2A); alternatively, anti-FtsZ drugs may hyperstabilize the binding of FtsZ polymers [5]. The best performing compound, derivative **11,** was docked in the crystal structure of the FtsZ. The two possible binding sites of **11** were identified as follows: (i) the GTP binding pocket (Figure 2B) and (ii) the hydrophobic cleft between the two FtsZ subdomains (Figure 2C) [19].

These docking poses were taken as starting conformations to carry out molecular dynamic (MD) simulations in explicit water, to verify the stability of the complexes and calculate the binding affinity of **11** in the two binding sites. The free energies of binding (Table 3), calculated applying the MM-GBSA methodology, showed that the binding of **11** in the interdomain cleft (∆G = −13.0 kcal mol^−1^) is strongly favored when compared to the GTP-binding pocket (∆G = 9.4 kcal mol^−1^).

The interaction of isatin-derivative **11** in the hydrophobic interdomain cleft is driven by the van der Waals (VDW) term, which accounts for −51.8 kcal mol^−1^ and comprises all the non-polar interactions between the protein residues and the ligand. The electrostatic (E_El_ = −2.6 kcal mol^−1^) and non-polar solvation (E_SURF_ = −6.1 kcal mol^−1^) terms are also favorable to the complex formation, while the polar solvation contribution (E_GB_ = 32.1 kcal mol^−1^) is detrimental to the binding. Of course, the entropic term opposes binding (∆S = 15.2 kcal mol^−1^). The per-residue decomposition analysis identifies the protein residues responsible for the binding of **11**; Gln181 and Thr289, through hydrogen bonding, and Gly185, Val286, and Ile189, through non-polar interactions, are the amino acids that mostly contribute to the binding (Figure 3).

Once we identified that **11** preferentially binds the interdomain cleft, we studied, by means of MD simulations, the effect of its binding in the polymerization of FtsZ. The simulations, followed by binding affinity calculations, suggested that the interaction between two apo-FtsZ is only transient, with a binding free energy of −3.4 kcal mol^−1^, a value close to zero (Figure 4A). By adding GTP and Ca^2+^, that are experimentally crucial, to promote the polymerization of FtsZ, the binding between two FtsZ proteins is strongly improved, reaching a value of −47.9 kcal mol^−1^ (Figure 4B). This result confirms that both GTP and divalent cations are necessary for the polymerization of FtsZ. Remarkably, the presence of **11** in the interdomain region of FtsZ knocks down the interaction between FtsZ proteins, with a positive binding free energy of +31.2 kcal mol^−1^ (Figure 4C), demonstrating the inhibitory activity of **11** toward the polymerization of FtsZ and explaining its antimicrobial activity.

Multiple sequence alignment of the primary sequences of FtsZ from *E. coli* (6LL6) with those of *S. aureus* (3VOB), *K. pneumoniae* (8GZV), *P. aeruginosa* (2VAW), *A. baumannii* (AF-B0VNZ4-F1), and *E. hormaechei* (AF-A0A0A5PLG7-F1), and the superimposition of their crystallographic structures suggest a high similarity between the proteins from different bacterial species (Appendix A). Thus, this result cannot explain the different behavior of the isatin derivatives against the two bacterial strains selected as model systems in the present study. A different explanation may be due to the difference between Gram-positive and Gram-negative cell wall architecture.

### 2.5. Combined Effect with Derivative ***11*** and Colistin

To shed light on the different potency of compound **11** against *S. aureus* and *E. coli* (Table 1), and the FtsZ protein being highly conserved in bacteria (Appendix A), a colistin association assay in *E. coli* was performed by combining the colistin, an outer membrane (OM) damaging antibiotic, at a sub-inhibitory concentration with derivative **11.** In this experimental condition in which the OM is made permeable to the diffusion of many molecules, derivative **11** turned active with a IC_50_ value of 54 μM, thus displaying its effectiveness while it allowed *E. coli* proliferation at regular extent when used at 100 μM without colistin (107.9 ± 2.4% of growth compared to the untreated control, see Table 1). This is confirmation of the ability of the isatin bis-imidathiazole hybrid to affect also Gram-negative by targeting FtsZ once it passes through the OM.

### 2.6. In Vitro Antibiofilm Evaluation

Having demonstrated the high antimicrobial potency of **11** against the planktonic form of *S. aureus*, together with its safety profile on mammalian cells, it was further investigated to determine its antibiofilm capacity in terms of inhibition of biofilm production (Figure 5). The biomass of *S. aureus* was quantitatively evaluated by means of a CV staining and results confirmed the effectiveness of derivative **11**. In the range 200–25 µM, the development of the biofilm was completely inhibited (<10% of biomass compared to the untreated control).

## 3. Conclusions

In the present study, a series of isatin bis-imidathiazole hybrids (**3**–**20**) was designed and synthetized by properly modifying isatin derivatives **6k** and **6m**, previously endowed with activity against *S. aureus* (IC_50_ values of 79.95 μM and 50.74 μM, respectively) [15]. The new set of derivatives was assayed in vitro against three priority human pathogens, *Staphylococcus aureus*, *Escherichia coli*, and *Candida albicans*; these microorganisms were selected as representative strains of Gram-positive and Gram-negative bacteria and a fungal model system, respectively. They are all causative agents of community- and hospital-acquired infections, characterized by high recurrence rates and increasing antimicrobial resistance to many classes of antimicrobial agents.

According to the microbiological evaluations, derivative **11** was the best performing isatin bis-imidathiazole hybrid in the series, being able to inhibit *S. aureus* proliferation with a low IC_50_ value (13.8 μM), thus displaying an improved potency compared to the two lead compounds (**6k** and **6m**). In addition, a strong antibacterial activity against multi-drug-resistant *S. aureus* isolates was observed, reducing the development of the biofilm at a non-cytotoxic concentration. Notably, its CC_50_ value on Vero cells was 63.3 μM, and it showed no hemolytic property even at the highest tested concentration.

Derivative **11** did not inhibit the growth of *E. coli* and *C. albicans*, but it partially restored its potency against the Gram-negative strain when tested in combination with the colistin. This finding suggests that the lack of activity is due to the outer membrane barrier and confirms how conserved FtsZ protein is in bacterial species (Appendix A). Not surprisingly, derivative **11** did not interfere with *C. albicans* proliferation because of the poor sequence homology between FtsZ in bacteria and tubulin in Eukarya [19].

Molecular docking, followed by MD simulations, identified the binding site of derivative **11** to target FtsZ, a key protein in bacterial cell division and the mechanism of action, i.e., inhibition of its polymerization. Notably, the active compound binds preferentially to the interdomain cleft of FtsZ rather than to the GTP binding site, thus ensuring selectivity of action on *S. aureus* strains due to the absence of homology with mammalian tubulin in this specific domain [19].

In summary, the overall data indicate that the isatin bis-imidathiazole hybrid **11**, bearing an OCF_3_ group at position 5 of the indolinone system, is a potent anti-staphylococcal agent targeting FtsZ protein; it appears to be effective on the reference strain of *S. aureus* as well as on multi-drug-resistant isolates and biofilms. In addition, for the compound, the ability to inhibit FtsZ polymerization was recognized as a mechanism of action through computational studies. Further studies should be performed to evaluate the potential development of drug resistance in susceptible strains, and to improve the antimicrobial spectrum of the new chemotype of isatin-based hybrids, following a molecular dynamics-assisted process design.

## 4. Materials and Methods

### 4.1. Chemistry

Thin layer chromatography was performed on Bakerflex plates (Silica gel IB2-F, Fisher Scientific, Waltham, MA, USA); the eluent was a mixture of petroleum ether/acetone in various proportions. The ^1^H NMR and ^13^C NMR spectra were recorded on a Varian MR 400 MHz (ATB PFG probe, Palo Alto, CA, USA) and on a Bruker ADVANCE NEO 600 MHz (Billerica, MA, USA), equipped with a liquid nitrogen chilled Bruker Prodigy (1H/19F)-X broadband probe.

The chemical shift was expressed in δ (ppm) and referenced to the residual peak of the solvent as the internal standard (DMSO-d_6_: δ H = 2.50 ppm; δ C = 39.52 ppm). The coupling constant values (*J*) were determined in Hertz (abbreviation: ph = phenyl, th = thiazol; ind = indole). UHPLC−MS analyses were run on a Waters ACQUITY ARC UHPLC/MS system (Milford, MA, USA), consisting of a QDa mass spectrometer equipped with an electrospray ionization interface and a 2489 UV/Vis detector at wavelengths (λ) 254 nm and 365 nm. Elemental analyses were within ± 0.4% of the theoretical values. Compounds were named relying on the naming algorithm developed by CambridgeSoft Corporation (Perkin Elmer, Milan, Italy) and used in Chem-BioDraw Ultra 14.0 (Perkin Elmer, Milan, Italy). All solvents and reagents, unless otherwise stated, were supplied by Aldrich Chemical Co. Ltd. (Milan, Italy) and 6-chloroimidazo[2,1-b]thiazole and 6-chloro-2-methylimidazo[2,1-b]thiazole (**1a**,**b**) has been prepared as described in the literature [20,21] The isatins, **2c**–**l**, were either commercially available or prepared as described in the literature [22,23,24,25,26,27] and were used without further purification. 

*Synthesis of New Compounds* **3**–**20**

Appropriately substituted isatin (1.72 mmol) was dissolved in 20 mL of isopropanol, taken in a reaction vessel, and iodine (5 mmol) was added to it. The appropriate imidazo[2,1-*b*]thiazole (3.46 mmol) was added to the mixture and the reaction was continued with constant stirring for 15–28 h (according to a TLC test). After completion of reaction, the mixture was concentrated under reduced pressure in a rotary evaporator. It was then extracted with dichloromethane, and the extract was washed with saturated sodium thiosulphate solution to decompose the remaining iodine. The organic solution was evaporated and the crude product was crystallized from ethanol.

*3,3-bis(6-Chloro-2-methylimidazo[2,1-b]thiazol-5-yl)-5-methoxyindolin-2-one* (**3**), Yield: 56%. ^1^H-NMR (DMSO-d_6_): δ 10.97 (s, 1H, NH), 6.99 (d, *J* = 2.4, 1H, ind), 6.94 (m, 2H, ind), 2.32 (d, *J* = 2.1, 3H, CH_3_), 6.85 (d, *J* = 2.1, 1H, th), 6.79 (d, *J* = 2.1, 1H, th), 3.65 (s, 3H, OCH_3_), 2.34 (d, *J* = 2.1, 3H, CH_3_). ^13^C-NMR (DMSO-d_6_): δ 172.54, 155.16, 146.65, 146.64, 134.61, 130.20, 129.39, 128.11, 127.11, 126.37, 116.55, 115.84, 115.79, 115.44, 114.67, 113.01, 111.32, 55.54, 50.78, 13.36. MS (ES): *m/z* calcd. for C_21_H_15_Cl_2_N_5_O_2_S_2_: 287.01 [M + H]^+^ 504.01; found 504.08. Anal. calcd. for C_21_H_15_Cl_2_N_5_O_2_S_2_ (MW 504.40): C, 50.01; H, 3.00; N, 13.88; found: 50.04; H, 2.99; N, 13.90.

*3,3-bis(6-Chloro-2-methylimidazo[2,1-b]thiazol-5-yl)-1-(4-chlorobenzyl)indolin-2-one* (**4**), Yield: 27%. ^1^H-NMR (DMSO-d_6_): δ 7.41 (m, 5H, 4ph + ind), 7.34 (d, *J* = 7.5, 1H, ind), 7.29 (d, *J* = 7.5, 1H, ind), 7.09 (t, *J* = 7.5, 1H, ind), 6.85 (d, *J* = 1.6, 1H, th), 6.61 (d, *J* = 1.6, 1H, th), 5.09 (d, *J* = 15.6, 1H, CH_2_), 5.00 (d, *J* = 15.6, 1H, CH_2_), 2.30 (d, *J* = 1.6, 3H, CH_3_), 2.23 (d, *J* = 1.6, 3H, CH_3_). ^13^C-NMR (DMSO-d_6_): δ 171.18, 146.88, 146.84, 141.73, 134.65, 132.55, 130.46, 130.35, 129.74, 129.49, 128.65, 127.18, 126.58, 126.06, 123.43, 116.45, 115.58, 115.21, 115.02, 110.42, 49.99, 43.07, 13.28, 13.16. MS (ES): *m/z* calcd. for C_27_H_18_Cl_3_N_5_OS_2_: [M + H]^+^ 600.01; found 598.10 Anal. calcd. for C_27_H_18_Cl_3_N_5_OS_2_ (MW 598.95): C, 54.14; H, 3.03; N, 11.69; found: C, 54.10; H, 3.02; N, 11.71.

*3,3-bis(6-Chloroimidazo[2,1-b]thiazol-5-yl)-5-fluoroindolin-2-one* (**5**), Yield: 35%. ^1^H-NMR (DMSO-d_6_): δ 11.25 (s, 1H, NH), 7.33 (d, *J* = 4.5, 1H, th), 7.28 (d, *J* = 4.5, 1H, th), 7.22 (td, *J* = 2.4, *J* = 8.5, 1H, ind-6), 7.10 (m, 2H, th + ind), 7.04 (dd, *J* = 4.4, *J* = 8.5, 1H, ind), 6.90 (d, *J* = 4.5, 1H, th). ^13^C-NMR (DMSO-d_6_): δ 172.65, 158.08 (d, *^1^J_C-F_* = 239.4 Hz), 148.06, 147.88, 137.80, 131.24, 130.35, 128.09 (d, *^3^J_C-F_* = 8.6 Hz), 119.38, 119.23, 116.87 (d, *^2^J_C-F_* = 22.2 Hz), 115.18 (d, *^2^J_C-F_* = 22.2 Hz), 114.93, 114.73, 113.83, 113.58, 111.87 (d, *^3^J_C-F_* = 8.6 Hz), 56.01. MS (ES): *m/z* calcd. for C_18_H_8_Cl_2_FN_5_OS_2_: [M + H]^+^ 463.96; found 463.98. Anal. calcd. for C_18_H_8_Cl_2_FN_5_OS_2_ (MW 464.31): C, 46.56; H, 1.74; N, 15.08; found: C, 46.58; H, 1.75; N, 15.10.

*3,3-bis(6-Chloro-2-methylimidazo[2,1-b]thiazol-5-yl)-5-fluoroindolin-2-one* (**6**), Yield: 34%. ^1^H-NMR (DMSO-d_6_): δ 11.16 (s, 1H, NH), 7.21 (td, *J* = 2.8, *J* = 8.8, 1H, ind), 7.07 (dd, *J* = 2.8, *J* = 8.8, 1H, ind), 7.03 (m, 1H, ind), 6.96 (d, *J* = 1.8, 1H, th), 6.84 (d, *J* = 1.8, 1H, th), 2.39 (d, *J* = 1.8, 3H, CH_3_), 2.31 (d, *J* = 1.8, 3H, CH_3_). ^13^C NMR (DMSO-d_6_): δ 172.62, 158.09 (d, *^1^J_C-F_* = 238.7 Hz), 146.93, 146.84, 137.70, 130.34, 129.38, 128.26 (d, *^3^J_C-F_* = 7.7 Hz), 127.21, 126.68, 116.80 (d, *^2^J_C-F_* = 24.5 Hz), 116.38, 115.87, 115.00, 114.97, 113.62 (d, *^2^J_C-F_* = 24.5 Hz), 111.89 (d, *^3^J_C-F_* = 7.7 Hz), 50.73, 13.34. MS (ES): *m/z* calcd. for C_20_H_12_Cl_2_FN_5_OS_2_: [M + H]^+^ 491.99; found 492.10. Anal. calcd. for C_20_H_12_Cl_2_FN_5_OS_2_ (MW 492.37): C, 48.79; H, 2.46; N, 14.22; found: C, 48.81; H, 2.45; N, 14.20.

*3,3-bis(6-Chloroimidazo[2,1-b]thiazol-5-yl)-6-fluoroindolin-2-one* (**7**), Yield: 11%. ^1^H-NMR (DMSO-d_6_): δ 11.35 (s, 1H, NH), 7.34 (d, *J* = 4.3, 1H, th), 7.28 (d, *J* = 4.3, 1H, th), 7.24 (m, 1H, ind), 7.12 (d, *J* = 4.3, 1H, th), 6.91 (1H, d, th, *J* = 4.3), 6.85 (2H, m, ind). ^13^C-NMR (DMSO-d_6_): δ 173.16, 163.31 (d, *^1^J_C-F_* = 245.4 Hz), 148.06 (d, *^2^J_C-F_* = 29.4 Hz), 143.19, 143.11, 131.34, 130.42, 127.93(d, *^3^J_C-F_* = 10.5 Hz), 122.67, 119.41 (d, *^2^J_C-F_* = 29.4 Hz), 115.62, 115.29 (d, *^3^J_C-F_* = 10.5 Hz), 114.90, 109.23, 109.08, 99.32, 99.14, 50.10. MS (ES): *m/z* calcd. for C_18_H_8_Cl_2_FN_5_OS_2_: [M + H]^+^ 463.96; found 464.16. Anal. calcd. for C_18_H_8_Cl_2_FN_5_OS_2_ (MW 464.31): C, 46.56; H, 1.74; N, 15.08; found: C, 46.58; H, 1.73; N, 15.10.

*3,3-bis(6-Chloro-2-methylimidazo[2,1-b]thiazol-5-yl)-6-fluoroindolin-2-one* (**8**), Yield: 10%. ^1^H NMR (DMSO-d_6_): δ 11.29 (s, 1H, NH), 7.23 (m, 1H, ind), 6.84 (m, 3H, 2 ind + th), 2.34 (d, *J* = 1.6, 3H, CH_3_), 2.32 (d, *J* = 1.6, 3H, CH_3_). ^13^C NMR (DMSO-d_6_): δ 173.02, 163.17 (d, *^1^J_C-F_* = 245.2 Hz), 146.86, 146.78, 143.00 (d, *^3^J_C-F_* = 11.4 Hz), 130.35, 129.41, 127.77 (d, *^3^J_C-F_* = 11.4 Hz), 127.21, 126.61, 122.71, 116.42, 115.84, 115.30, 115.24, 108.95 (d, *^2^J_C-F_* = 25.1 Hz), 99.07 (d, *^2^J_C-F_* = 25.1 Hz), 49.99, 13.34. MS (ES): *m/z* calcd. for C_20_H_12_Cl_2_FN_5_OS_2_: [M + H]^+^ 491.99; found 492.17. Anal. calcd. for C_20_H_12_Cl_2_FN_5_OS_2_ (MW 492.37): C, 48.79; H, 2.46; N, 14.22; found: C, 48.81; H, 2.45; N, 14.20.

*3,3-bis(6-Chloroimidazo[2,1-b]thiazol-5-yl)-6-iodoindolin-2-one* (**9**), Yield: 25%. ^1^H-NMR (DMSO-d_6_): δ 11.26 (s, 1H, NH), 7.39 (dd, *J* = 1.5, *J* = 8.1, 1H, ind), 7.35 (d, *J* = 1.5, 1H, ind), 7.33 (d, *J* = 4.6, 1H, th), 7.28 (d, *J* = 4.6, 1H, th), 7.12 (d, *J* = 4.6, 1H, th), 7.03 (d, *J* = 8.1, 1H, ind), 6.91 (d, *J* = 4.6, 1H, th). ^13^C-NMR (DMSO-d_6_): δ 172.39, 148.05, 147.85, 142.80, 131.26, 131.20, 130.34, 128.01, 126.36, 119.39, 119.24, 119.19, 115.13, 114.87, 114.77, 95.84, 50.24. MS (ES): *m/z* calcd. for C_18_H_8_Cl_2_IN_5_OS_2_: [M + H]^+^ 571.87; found 571.99. Anal. calcd. for C_18_H_8_Cl_2_IN_5_OS_2_ (MW 572.22): C, 37.78; H, 1.41; N, 12.24; found: C, 37.80; H, 1.41; N, 12.22.

*3,3-bis(6-Chloro-2-methylimidazo[2,1-b]thiazol-5-yl)-6-iodoindolin-2-one* (**10**), Yield: 37%. ^1^H-NMR (DMSO-d_6_): δ 11.22 (s, 1H, NH), 7.39 (dd, *J* = 1.2, *J* = 7.8, 1H, ind), 7.34 (d, *J* = 1.2, 1H, ind), 7.00 (d, *J* = 7.8, 1H, ind), 6.96 (d, *J* = 1.2, 1H, th), 6.88 (d, *J* = 1.2, 1H, th), 2.34 (d, *J* = 1.2, 3H, CH_3_), 2.32 (d, *J* = 1.2, 3H, CH_3_). ^13^C NMR (DMSO-d_6_): δ 172.42, 146.90, 146.82, 142.77, 131.15, 130.42, 128.03, 127.27, 126.60, 126.48, 119.26, 116.47, 115.82, 114.92, 114.80, 95.76, 50.24, 13.34. MS (ES): *m/z* calcd. for C_20_H_12_Cl_2_IN_5_OS_2_: [M + H]^+^ 599.90; found 600.10. Anal. calcd. for C_20_H_12_Cl_2_IN_5_OS_2_ (MW 600.27): C, 40.02; H, 2.02; N, 11.67; found: C, 40.05; H, 2.01; N, 11.65.

*3,3-bis(6-Chloroimidazo[2,1-b]thiazol-5-yl)-6-(trifluoromethoxy)indolin-2-one* (**11**), Yield: 9%. ^1^H-NMR (DMSO-d_6_): δ 11.34 (s, 1H, NH), 7.35 (m, 2H, ind + th), 7.29 (d, *J* = 4.7, 1H, th), 7.13 (d, *J* = 4.7, 1H, th), 7.00 (m, 2H, ind), 6.92 (d, *J* = 4.7, 1H, th). ^13^C-NMR (DMSO-d_6_): δ 172.76, 149.48, 148.17, 147.95, 143.07, 131.34, 130.31, 127.84, 125.62, 119.35, 119.24, 118.72, 115.22, 114.91, 114.79, 114.70, 103.88, 50.07. MS (ES): *m/z* calcd. for C_19_H_8_Cl_2_F_3_N_5_O_2_S_2_: [M + H]^+^ 529.95; found 530.01. Anal. calcd. for C_19_H_8_Cl_2_F_3_N_5_O_2_S_2_ (MW 530.32): C, 43.03; H, 1.52; N, 13.21; found: C, 42.99; H, 1.53; N, 13.23.

*3,3-bis(6-Chloro-2-methylimidazo[2,1-b]thiazol-5-yl)-6-(trifluoromethoxy)indolin-2-one* (**12**), Yield: 15%. ^1^H-NMR (DMSO-d_6_): δ 11.38 (s, 1H, NH), 7.32 (d, *J* = 8, 1H, ind), 7.00 (m, 3H, 2 ind + th), 6.87 (d, *J* = 1.5, 1H, th), 2.34 (d, *J* = 1.5, 3H, CH_3_), 2.32 (d, *J* = 1.5, 3H, CH_3_). ^13^C-NMR (DMSO-d_6_): δ 172.78, 149.48, 149.47, 147.01, 146.91, 143.12, 130.48, 129.36, 127.29, 127.33, 126.76, 125.82, 121.27, 116.40, 115.91, 114.95, 114.86, 114.66, 103.90, 50.06, 13.36. MS (ES): *m/z* calcd. for C_21_H_12_Cl_2_F_3_N_5_O_2_S_2_: [M + H]^+^ 557.98; found 558.00. Anal. calcd. for C_21_H_12_Cl_2_F_3_N_5_O_2_S_2_ (MW 558.38): C, 45.17; H, 2.17; N, 12.54; found: C, 45.20; H, 2.16; N, 12.52.

*1-Benzyl-3,3-bis(6-chloroimidazo[2,1-b]thiazol-5-yl)indolin-2-one* (**13**), Yield: 25%. ^1^H-NMR (DMSO-d_6_): δ 7.40 (td, *J* = 1.2, *J* = 7.6, 1H, ind), 7.28 (m, 9H, 2 th + 2 ind + 5 ph), 7.09 (td, *J* = 1.2, *J* = 7.6, 1H, ind), 6.97 (d, *J* = 4.8, 1H, th), 6.88 (d, *J* = 4.8, 1H, th), 5.09 (d, *J* = 15.6, 1H, CH_2_), 5.02 (d, *J* = 15.6, 1H, CH_2_). ^13^C-NMR (DMSO-d_6_): δ 171.15, 148.02, 147.89, 141.98, 135.51, 131.25, 130.36, 130.32, 128.65, 127.73, 127.35, 126.01, 125.84, 123.37, 119.23, 119.19, 115.58, 115.19, 114.91, 114.85, 110.53, 50.00, 43.61.MS (ES): *m/z* calcd. for C_25_H_15_Cl_2_N_5_OS_2_: [M + H]^+^ 536.02; found 536.09. Anal. calcd. for C_25_H_15_Cl_2_N_5_OS_2_ (MW 536.45): C, 55.97; H, 2.82; N, 13.06; found: C, 56.00; H, 2.82; N, 13.04.

*1-Benzyl-3,3-bis(6-chloro-2-methylimidazo[2,1-b]thiazol-5-yl)indolin-2-one* (**14**), Yield: 37%. ^1^H-NMR (DMSO-d_6_): δ 7.41 (td, 1H, *J* = 1.3, *J* = 7.7, ind), 7.31 (m, 7H, 2 ind + 5 ph), 7.09 (td, 1H, *J* = 1.3, *J* = 7.7, ind), 6.88 (d, *J* = 1.3, 1H, th), 6.63 (th, *J* = 1.3, 1H, d), 5.10 (d, *J* = 15.6, 1H, CH_2_), 5.00 (d, *J* = 15.6, 1H, CH_2_), 2.31 (d, *J* = 1.3, 3H, CH_3_), 2.22 (d, *J* = 1.3, 3H, CH_3_). ^13^C-NMR (DMSO-d_6_): δ 171.23, 146.90, 146.85, 141.98, 135.64, 130.50, 130.37, 129.49, 128.74, 127.89, 127.70, 127.23, 126.54, 126.13, 126.05, 123.38, 116.63, 115.63, 115.35, 115.06, 110.52, 50.05, 43.72, 13.34, 13.29. MS (ES): *m/z* calcd. for C_27_H_19_Cl_2_N_5_OS_2_: [M + H]^+^ 564.05; found 564.20. Anal. calcd. for C_27_H_19_Cl_2_N_5_OS_2_ (MW 564.50): C, 57.45; H, 3.39; N, 12.41; found: C, 57.48; H, 3.40; N, 12.39.

*3,3-bis(6-Chloroimidazo[2,1-b]thiazol-5-yl)-1-cinnamylindolin-2-one* (**15**), Yield: 45%. ^1^H-NMR (DMSO-d_6_): δ 7.44 (t, *J* = 7.6, 1H, ind), 7.31 (8H, m, 2th + 2ind + 4ph), 7.24 (t, *J* = 6, 1H, ph), 7.13 (d, *J* = 4.5, 1H, th), 7.10 (t, *J* = 7.6, 1H, ind), 6.92 (d, *J* = 4.5, 1H, th), 6.48 (d, 1H, CH, *J* =16.2), 6.38 (dt, *J* = 5.4, *J* = 16.5, 1H, CH), 4.66 (ddd, *J* =1.5, *J* = 5.4, *J* = 16.5, 1H, CH_2_), 4,60 (ddd, *J* = 1.5, *J* = 5.4, *J* = 16.5, 1H, CH_2_). ^13^C-NMR (DMSO-d_6_): 170.80, 148.05, 147.89, 141.94, 135.80, 131.40, 131.23, 130.37, 130.32, 128.71, 127.88, 126.20, 125.95, 125.83, 123.31, 122.53, 119.36, 119.28, 115.67, 115.21, 114.99, 114.89, 110.52, 50.03, 41.72. MS (ES): *m/z* calcd. for C_27_H_17_Cl_2_N_5_OS_2_: [M + H]^+^ 562.03; found 562.19. Anal. calcd. for C_27_H_17_Cl_2_N_5_OS_2_ (MW 562.49): C, 57.65; H, 3.05; N, 12.45; found: C, 57.68; H, 3.05; N, 12.46.

*3,3-bis(6-Chloro-2-methylimidazo[2,1-b]thiazol-5-yl)-1-cinnamylindolin-2-one* (**16**), Yield: 52%. ^1^H-NMR (DMSO-d_6_): δ 7.44 (t, *J* = 7.7, 1H, ind), 7.35 (d, *J* = 7.7, 1H, ind), 7.31 (m, 5H, ph), 7.24 (m, 1H, ind), 7.11 (t, *J* = 7.7, 1H, ind), 6.90 (s, 2H, th), 6.48 (d, *J* = 16.2, 1H, CH), 6.42 (dt, *J* = 5.2, *J* = 16.2, 1H, CH), 4.67 (dd, *J* = 5.2, *J* =16.2, 1H, CH_2_), 4.60 (dd, *J* = 5.2, *J* = 16.2, 1H, CH_2_), 2.31 (d, *J* = 1.2, 3H, CH_3_), 2.21 (d, *J* = 1.2, 3H, CH_3_). ^13^C-NMR (DMSO-d_6_): δ 170.86, 146.85, 141.87, 135.75, 131.67, 130.41, 130.35, 129.42, 128.88, 128.72, 128.19, 127.93, 127.15, 126.57, 126.22, 126.02, 125.99, 123.27, 122.41, 116.53, 115.90, 115.34, 115.21, 110.45, 50.04, 41.72, 13.31, 13.14. MS (ES): *m/z* calcd. for C_29_H_21_Cl_2_N_5_OS_2_: [M + H]^+^ 590.06; found 590.19. Anal. calcd. for C_29_H_21_Cl_2_N_5_OS_2_ (MW 590.54): C, 58.98; H, 3.58; N, 11.86; found: C, 59.00; H, 3.58; N, 11.85.

*3-(6-Chloroimidazo[2,1-b]thiazol-5-yl)-3-hydroxy-4-(trifluoromethyl)indolin-2-one* (**17**), Yield: 3%. ^1^H-NMR (DMSO-d_6_): δ 10.94 (s, 1H, NH), 8.20 (d, *J* = 4.5,1H, th), 7.55 (t, *J* = 8.1, 1H, ind), 7.46 (s, 1H, OH), 7.36 (d, 1H, *J* = 4.5, th), 7.30 (d, *J* = 8.1, 1H, ind), 7.19 (d, *J* = 8.1, 1H, ind). ^13^C-NMR (DMSO-d_6_): δ 175.04, 146.38, 144.26, 131.89, 131.13, 127.85, 122.36, 117.78, 114.50, 112.62, 109.92, 93.77, 75.71, 54.91. MS (ES): *m/z* calcd. for C_14_H_7_ClF_3_N_3_O_2_S: [M + H]^+^ 374.00; found 374.20. Anal. calcd. for C_14_H_7_ClF_3_N_3_O_2_S (MW 373.73): C, 44.99; H, 1.89; N, 11.24; found: C, 45.01; H, 1.90; N, 11.23.

*3-(6-Chloro-2-methylimidazo[2,1-b]thiazol-5-yl)-3-hydroxy-4-(trifluoromethyl)indolin-2-one* (**18**), Yield: 57%. ^1^H-NMR (DMSO-d_6_): δ 10.86 (s, 1H, NH), 7.54 (t, *J* = 7.8, 1H, ind), 7.42 (s, 1H, th), 7.30 (d, *J* = 7.8, 1H, ind), 7.19 (d, *J* = 7.8, 1H, ind), 2.45 (d, *J* = 1.8, 3H, CH_3_). ^13^C-NMR (DMSO-d_6_): δ 175.23, 144.76 (d, *^1^J_C-F_* = 234 Hz), 131.47, 127.18, 126.88(d, *^2^J_C-F_* = 19.6 Hz), 126.53, 124.92, 124.46, 122.65, 119.39(d, *^4^J_C-F_* = 4.5 Hz), 119.10, 118.76, 114.29, 73.91, 13.38. MS (ES): *m/z* calcd. for C_15_H_9_ClF_3_N_3_O_2_S: [M + H]^+^ 388.01; found 388.14. Anal. calcd. for C_15_H_9_ClF_3_N_3_O_2_S (MW 387.76): C, 46.46; H, 2.34; N, 10.84; found: C, 46.48; H, 2.34; N, 10.85.

*3-(6-Chloroimidazo[2,1-b]thiazol-5-yl)-3-hydroxy-4-iodoindolin-2-one* (**19**), Yield: 40%. ^1^H- NMR (DMSO-d_6_): δ 10.69 (s, 1H, NH), 8.21 (s, 1H, OH), 7.36 (m, 2H, th), 7.24 (s, 1H, ind), 7.05 (t, *J* = 7.8, 1H, ind), 6.90 (d, *J* = 7.8, 1H, ind). ^13^C-NMR (DMSO-d_6_): δ 175.04, 146.38, 144.26, 131.89, 131.13, 127.85, 122.36, 117.78, 112.62, 109.92, 93.77, 75.71, 54.91. MS (ES): *m/z* calcd. for C_13_H_7_ClIN_3_O_2_S: [M + H]^+^ 431.91; found 431.97 Anal. calcd. for C_13_H_7_ClIN_3_O_2_S (MW 431.63): C, 36.17; H, 1.63; N, 9.74; found: C, 36.20; H, 1.63; N, 9.76.

*3-(6-Chloro-2-methylimidazo[2,1-b]thiazol-5-yl)-3-hydroxy-4-iodoindolin-2-one* (**20**), Yield: 25%. ^1^H-NMR (DMSO-d_6_): δ 10.75 (s, 1H, NH), 8.03 (s, 1H, OH), 7.37 (d, *J* = 7.8, 1H, ind), 7.20 (d, *J* = 1.2, 1H, th), 7.05 (t, *J* = 7.8, 1H, ind), 6.89 (d, *J* = 7.8, 1H, ind), 2.46 (d, *J* = 1.2, 3H, CH_3_). ^13^C-NMR (DMSO-d_6_): δ 175.11, 145.45, 144.31, 131.94, 131.89, 131.18, 126.86, 124.84, 119.26, 117.63, 109.94, 93.93, 75.70, 13.43. MS (ES): *m/z* calcd. for C_14_H_9_ClIN_3_O_2_S: [M + H]^+^ 445.92; found 445.96. Anal. calcd. for C_14_H_9_ClIN_3_O_2_S (MW 445.66): C, 37.73; H, 2.04; N, 9.43; found: C, 37.71; H, 2.05; N, 9.45.

### 4.2. Microbial Strains and Growth Conditions

Reference bacterial strains of *Staphylococcus aureus* (ATCC 25923), *Escherichia coli* (ATCC 25922), and *Candida albicans* (ATCC 10231) were used in this study as Gram-positive, Gram-negative, and fungal models, respectively. These reference strains were purchased from the American Type Culture Collection (ATCC, Manassas, VA, USA); bacterial cultures were routinely grown at 37 °C on 5% blood agar plate, while fungal cultures on Sabouraud Dextrose Agar (Biolife Italiana S.r.l., Milan, Italy).

### 4.3. Compounds and Reference Drugs

The dry powder of the isatin derivatives was resuspended in dimethylsulfoxide (DMSO) at 20 mM and used as stock solutions. For biological investigations, compounds were used in the range 100–0.78 µM, and the solvent in the corresponding percentage range.

The following commercially available antimicrobials and toxic clinical drug were purchased from Sigma-Aldrich (St. Louis, MO, USA), dissolved in water, and stored at 4 °C. The gentamicin and the ampicillin were used as reference controls in the antimicrobial testing with *S. aureus* and *E. coli* in the range 107.69–0.82 µM and 143.23–1.12 µM, respectively. The colistin was used at 0.03 µM in association with some selected compounds with *E. coli* [28]. The fluconazole was used in the assays with *C. albicans* in the range 3.27–0.03 µM. The doxorubicin was included as clinical drug control in the cytotoxicity studies with Vero cells (ATCC CCL-81) in the range 367.97–0.72 µM.

### 4.4. Antimicrobial Activity

The antimicrobial activity of the compounds was assessed by a well-established broth microdilution procedure in microtiter plates and in compliance with the Clinical and laboratory Standard Institute (CLSI) guidelines (Clinical and Laboratory Standards Institute, Wayne, PA, USA). In short, microbial inocula were prepared at 0.5 McFarland in PBS and, subsequently, bacterial suspensions were diluted 1:200 in Mueller–Hinton broth (Sigma-Aldrich, St. Louis, MO, USA), while fungal inoculum was diluted 1:20 in RPMI-1640 medium (Gibco®, ThermoFisher Scientific Inc., Waltham, MA, USA), containing 2% glucose and 0.3% levo-glutamine buffered to pH 7.0 with 0.165 M 3-(N-morpholino)propanesulfonic acid (MOPS). A total of 100 µL of these microbial suspensions were introduced in a 96-well microplate and incubated with 100 µL of the compounds, two-fold serially diluted in the range of 100–0.19 µM. Positive controls (microbial suspensions in regular media), negative controls (only compounds), and solvent controls (microbial suspensions incubated with DMSO dilutions) were included in the tests. The plate was incubated at 37 °C for 24 h, and subsequently, the optical density at 630 nm (OD_630nm_) was spectrophotometrically measured. Each experiment was carried out with three technical replicates (i.e., three wells per sample) and repeated at least two times for statistical power. Compounds demonstrating inhibitory activity at 100 µM were further assayed in the range 100–0.78 µM in order to obtain dose–response curves. IC_50_ values, corresponding to the concentration of the compound giving rise to an inhibition of growth of 50%, were obtained by interpolation on dose–response curves generated by plotting the percentages of growth inhibition, relative to the positive control (set to 100% of growth), as a function of the tested concentrations (on a logarithm scale). Statistical analysis was carried out using GraphPad Prism version 9.4.1 for Windows (GraphPad Software, San Diego, CA, USA, www.graphpad.com).

### 4.5. Antibiofilm Activity

For the evaluation of the bacterial biofilm formation, an established protocol with minor modifications was used [29]. In short, a bacterial suspension of *S. aureus* ATCC 25923 was prepared in TSB (Tryptic Soy broth (TSB, Sigma-Aldrich, St. Louis, MO, USA) supplemented with 1% of glucose and adjusted to a final density of 10^5^ CFU/mL. Aliquots of 100 µL of the culture were transferred to a 96-well flat-bottom polystyrene microplate, and then incubated at 37 °C for 90 min to promote bacterial adhesion. Thereafter, wells were slowly rinsed with PBS to remove non-attached cells, and 100 µL of medium containing the 2-fold serial dilutions of the compounds were added. After incubation at 37 °C for 24 h, the free-floating cells were removed, and the biofilms were carefully washed with PBS, and incubated for 1 h at 60 °C. Bacterial biomass was quantified by a standardized crystal violet (CV) staining. Briefly, 100 µL of CV solution (0.1% in water) were added to each well containing a completely air-dried biofilm, and incubated for 30 min at 37 °C. Then, wells were washed twice with water to remove the unbound dye, and CV was dissolved with 100 µL of 95% ethanol for 30 min. Finally, the colored supernatants were transferred to a new microplate and the OD_550nm_ was read. The inhibitory effect of the compounds on biofilm production was evaluated in comparison to the positive control (*S. aureus* grown in absence of compounds), and negative control (only medium).

### 4.6. Cell Viability and Proliferation Assay

Vero cell line (ATCC CCL-81) was selected as the model system to assess the effect of the isatin hybrids on normal epithelial cells, and a previously established protocol was used with minor modifications [30]. Briefly, cells were cultured in RPMI-1640 medium supplemented with 10% fetal bovine serum (FBS) (Carlo Erba Reagents, Milan, Italy), 100 U/mL penicillin, and 100 µg/mL streptomycin at 37 °C with 5% CO_2_. For experiments, cells were seeded into 96-well plates at 10^4^ cells/well, and incubated at 37 °C for 24 h. Following washes with PBS, the cell monolayer was incubated with 100 µL of medium containing the 2-fold serial dilutions of the compounds. Both untreated cells and cells incubated with medium containing solvent dilutions were included in each experiment as controls. The cell viability was assessed by a WST8-based assay according to the manufacturer’s instructions (CCK-8, Cell Counting Kit-8, Dojindo Molecular Technologies, Rockville, MD, USA). After 48 h of incubation, culture medium was removed from each well, the monolayer was washed with PBS, and 100 µL of fresh medium containing 10 µL of CCK-8 solution were added. After 2 h at 37 °C, the OD_450nm_ was read, and data were expressed as the percentage of the cell viability relative to the untreated controls. The CC_50_ was obtained from the corresponding dose–response curves generated, as previously reported for IC_50_ values. 

### 4.7. Hemolytic Assay

The hemolytic activity of some selected compounds was evaluated as the amount of hemoglobin released by the disruption of human red blood cells (hRBCs) [18]. For the experiments, fresh hRBCs, obtained from peripheral blood of anonymous blood donors available for research purposes, were collected by centrifugation at 1500 g for 10 min, washed 3 times with PBS, and resuspended to a final concentration of 4% *w/v* hRBCs in PBS. Then, 100 µL of hRBCs suspension and an equal volume of the 2-fold dilutions of the antimicrobial peptides were mixed in a 96-well plate and incubated for 1 h at 37 °C. After centrifugation at 1000 g for 5 min, the supernatants were transferred into a clear 96-well plate; if compounds caused hemolysis, hemoglobin (along with other cellular constituents) was released in the supernatants and, as hemoglobin has a distinct absorbance spectrum, the degree of hemolysis in solution was measured using a spectrophotometer at 405 nm [18]. Untreated hRBCs (incubated with PBS) and hRBCs incubated with 1% Triton X-100 were employed as negative and positive controls, respectively. The hemolysis percentage was calculated as [OD_405nm_ (sample) − OD_405nm_ (negative control)]/[OD_405nm_ (positive control) − OD_405nm_ (negative control)] × 100. Minimal hemolytic concentrations (MHCs) were defined as the compound concentrations causing 10% hemolysis.

### 4.8. Statistical Analysis

Statistical analyses were carried out using Prism8 software (GraphPad 8.0, San Diego, CA, USA). Dose–response curves were obtained from nonlinear regressions and using the equation “log(inhibitor) vs. normalized response–variable slope”. Microbial and cellular determinations were analyzed with the two-ANOVA test. The in vitro experiments were performed in three independent biological replicates. A value of *p* < 0.05 was considered significant.

### 4.9. Computational Details

The crystal structure of FtsZ from *Staphylococcus aureus* (PDB ID 3VOB) was selected as the model protein for the computational investigations [31]. The protein was modeled using the Amber ff14SB force field [32], while guanosine-5′-triphosphate (GTP) [33] and compound **11** were parameterized using GAFF (Generalized Amber Force Field). The atomic charges of GTP and **11** were obtained following an optimization at HF/6-31G* level of theory, through Gaussian16 [34], and the application of the RESP (Restraint Electrostatical Potential Method) fitting approach [35].

Docking of derivative **11** in FtsZ was carried out using AutoDock Vina [36]. A total of 3 runs were carried out using exhaustiveness equal to 32 and an energy range of 100.

Explicit solvent molecules (TIP3P water model) and counterions were added to the two docking poses and then minimized by 10000 steps (5000 via steepest descend and 5000 via conjugate gradient algorithms). An equilibration step (1 ns) was then applied to heat the system up to 300 K (Langevin thermostat). MD simulations were carried out for 100 ns in Periodic Boundary Conditions (PBC), along with the particle mesh Ewald summations (cut-off radius of 10 Å). Amber22 was employed to run the MD simulations [37]. Molecular Mechanics–Generalized Born Surface Area (MM–GBSA) analysis was used to compute the binding affinity of **11** for FtsZ in the two binding pockets identified by the docking protocol [38]. Normal modes analysis was carried out to compute the entropy contribution to the binding. Post-processing analysis of the MD trajectories, such as root of mean squared deviation (RMSD), root of mean squared fluctuation (RMSF), gyration radius (RG), Solvent Accessible Surface Area (SASA), and hydrogen bon analysis were carried out employing CPPTRAJ and the results are illustrated in the Appendix A.

## Data Availability

The data presented in this study are available in this article.

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
