# Peer review of "Isatin Bis-Imidathiazole Hybrids Identified as FtsZ Inhibitors with On-Target Activity Against *Staphylococcus aureus"

_antibiotics, 2024, doi:10.3390/antibiotics13100992_

Round 1

Reviewer 1 Report

Comments and Suggestions for Authors

Comments on the Quality of English Language

Overall quality of English language is good. Minor corrections are needed as detailed in the attached file.

Author Response

Point-by-point letter of answers to the Reviewer 1’s comments

 Brief Summary

The authors synthesized a set of 18 derivatives of isatin as FtsZ inhibitors and studied their antimicrobial activities predominantly against S. aureus. Out of these 18 compounds, one derivative showed promising antimicrobial activity against S. aureus. The authors studied the toxicity of this lead compound in mammalian epithelial cells and in human RBCs. The authors used docking and molecular dynamics simulations to determine the binding site of the lead compound in target protein FtsZ.

General concept comments

The authors did not address the design principles behind these isatin derivatives, nor did they thoroughly investigate or explain the observed structure-activity relationships (SAR). For instance, there is no explanation for why introducing a methyl group at the R position in compound 12 completely eliminated its antimicrobial activity compared to compound 11. Similarly, the lack of antimicrobial activity of compound 11 against Candida albicans was not explained. In general, more detailed experimental design and thorough explanations of the findings would have significantly improved the quality of the article.

We thank the referee for the comments. However, we would like to point out that in the Introduction section four references well document the potential role of the heterocyclic compounds as FtsZ inhibitors [8-11]. In particular, ref. 11 (Lian, Z.-M.; Sun, J.; Zhu, H.-L. Design, Synthesis and Antibacterial Activity of Isatin Derivatives as FtsZ Inhibitors. J. Mol. Struct. 2016, 1117, 8–16, doi:10.1016/j.molstruc.2016.03.036) reports the antibacterial activity of isatin derivatives. Thus, considering that we previously identified isatin-hybrids (6k and 6m) endowed with excellent inhibitory activity against Staphylococcus aureus [ref. 15], the present paper continues our study on the rational design of isatin bis-imidazothiazole hybrids inspired to these lead compounds and adds knowledge on the possible mechanism of action of our best-performing analogue (11). As reported in the Conclusion section, further studies will be performed to delve into the antimicrobial potential of the isatin-based hybrids following a molecular dynamics-assisted process design. In this frame, computational methodology will allow for a deep understanding of the binding interaction of derivatives and FtsZ protein, including those who lack activity (12).

As for the lack of antifungal properties of compound 11, the observed resistance of Candida albicans is likely due to the poor sequence homology between bacterial FstZ protein and tubulin in Eukarya. Despite structural and functional similarities, FtsZ is a distant ancestral homolog of tubulin with an amino acid sequence that is <20% identical. For this reason, FtsZ protein is regarded as a specific target in bacteria. A comment in this regard has been added in the Conclusion section.

Specific comments for Abstract

  1. In line #22, the authors wrote “In the present study a library…” – I would prefer the term “series” instead of “library” to represent a set of 18 compounds. Please update it throughout the article.

The term “library” was replaced with “series” throughout the text.

  1. In line #26, the authors wrote “in absence of toxicity towards mammalian fibroblasts…” – I did not find any toxicity data on mammalian fibroblasts. Vero cells (ATCC CCL-81) are kidney epithelial cells.

It was a mistake. The abstract and the text were modified accordingly.

Specific comments for Introduction

  1. In line #50 and #51, “cytoscheleton” and “cytoscheletal” should be cytoskeleton and cytoskeletal, respectively.

Done.

  1. In line #53-54, the authors wrote “Indeed, FtsZ is an ideal drug target due to the highly conserved protein found in bacteria…” – the authors have only provided FtsZ sequence alignment data for E. coli and S. aureus. Showing the comparison between the FtsZ sequences of other multi-drug-resistant infectious microorganisms (e.g., C. albicans, K. pneumoniae, P. aeruginosa, A. baumannii, Enterobacter spp.) would strengthen the claim. I also recommend author to add a cross-reference of Table S1 after this statement.

We followed the suggestion of the referee and in the revised version of the manuscript we provided FtsZ sequence alignment data for E. coli, S. aureus, K. pneumoniae, P. aeruginosa, A. baumannii and Enterobacter spp. C. albicans is a fungus and not a bacterium (see also the response to General concept comments). We also added a cross-reference of this Figure S2 in the main text.

  1. In line #55, “β” in “β-tubulin” should be formatted correctly.

Done.

Specific comments for Results and discussion

  1. In line #101-102, the authors wrote “However, only the monoderivatives were obtained in this case too (Scheme 1).” – I recommend adding the structures of the monoderivatives in the supplementary information and provide a cross-reference after this statement.

We thank the referee for the comment. However, we would like to point out that the structures of the monosubstituted derivatives (17-20) are already reported in Scheme 1 (see also the following point).

  1. There is no table number where the isatin derivatives were listed. This should be Table 1. Please update the remaining tables and cross-references accordingly. Also, I recommend starting the table on the next page or repeating the header row on the next page.

We thank the referee for the comment. The substituents of the compounds are already reported in Scheme 1 which comprises also the table. To better clarify this, we have moved the caption of Scheme 1 before it and we have put Scheme 1 in a single page.

  1. In line #133-134, the authors wrote “the overall inhibitory effect of the compounds at 100 µM on the different model systems is reported in Table 1” – I believe the authors are reporting the percentages of viable microbial cells after treatment with 100 µM compound. I recommend authors to rewrite this sentence for clarity. Please update the title of Table 1 as well.

The referee is right. The sentence was rephrased.

  1. In line #137, the authors wrote “1 Previously reported in ref [15].” – It is unclear which compounds were reported previously in reference #15.

Table 1 was modified by changing the names for compounds 1 and 2 with 6k and 6m, respectively. These derivatives are the lead compounds previously described in ref. 15, used for the development of the new series. The apex was added in the table as well as the footnote.

  1. It is unclear in line #149-150, what cells were used to determine the CC50 values.

The text and Table 2 were modified to specify the model system used for CC50 determination.

  1. In line #176 “antibiotic profile” should be “antibiotic resistance profile”.

Done.

  1. Please provide a reference for the statement in line #195-196 “Anti-FtsZ drugs block FtsZ polymerization impeding the binding of the critical GTP cofactor or obstructing FtsZ subunit interactions sites…”

Reference 5 is reported at the end of the sentence.

  1. In line #228-229, the authors wrote “…we studied the effect of its binding in the polymerization of FstZ.” The authors should clarify that they used MD simulations to study the FtsZ polymerization in this sentence. Also, in line #229, it should be FtsZ not FstZ.

The referee is right. It was a typo.

  1. In line #234-235, the authors wrote “This result confirms that both GTP and divalent cations are necessary for the polymerization of FstZ.” – this is a known fact for actin/tubulin type of proteins.

Computational data confirmed this experimental evidence, because the binding between two FtsZ proteins was strongly improved in presence of GTP and divalent cations, passing from -3.4 kcal mol-1 to -47.9 kcal mol-1.

  1. In line #249-250, the authors wrote “A different explanation can be related to the different uptake of the compounds.” – I recommend adding “due to the difference between grampositive and gram-negative cell wall architecture.” for better understanding.

Done.

  1. In line #251, I recommend renaming the section heading from “Colistin association assay” to “Combined effect with derivative 11 and Colistin” or similar.

Done.

  1. In line #257, the authors reported an “IC50 value of 54 µM” in presence of colistin. I recommend reporting the IC50 of derivative 11 in E. coli without colistin?

IC50 value of compound 11 cannot be measured in E. coli as it did not display inhibitory activity at the highest tested concetration. However, to emphasize the different potency of the compound with and without colisitin, we added the percentage value of E. coli proliferation when used at 100 mM.

Specific comments for Conclusion

  1. Conclusion in line #274 should be paragraph #3. Materials and Methods in line #306 should be paragraph #4.

Done.

  1. In line #276-277, the authors wrote “…6k and 6m [15] as the head framework to obtain antimicrobial agents with improved potency.” – I recommend reporting the IC50 values of 6k and 6m against S. aureus and E. coli for better comparison.

Done.

  1. In line # 278-279, the authors wrote “The antimicrobial effect was assayed in vitro against three priority human pathogens, Staphylococcus aureus, Escherichia coli and Candida albicans.” – the authors did not evaluate all compounds thoroughly for all three pathogens. While it is not necessary to evaluate all compounds thoroughly, the authors did neither report IC50 value of compound 11 against Candida albicans nor explained why compound 11 failed to show any activity in Candida albicans. I recommend either adding more data or rephrasing this sentence to reflect what was actually done in this study.

We thank the referee for the comment. However, we would like to point out that all the compounds were assayed against the three microorganisms at 100 mM, and results are reported in Table 1. We specified the pipeline of the study in Paragraph 2.2. for a better understanding of the experimental design.

As for the IC50 values, dose-response curves can be produced only for the active compounds, thus IC50 values is not measurable for compound 11 that failed to inhibit C. albicans (percentage of proliferation of 107 % at 100 mM). This finding is due to the poor sequence homology between bacterial FtsZ protein and tubulin in Eukarya.

  1. In line #301-302, the authors wrote “…FtsZ ligand with an on-target…” - This cannot be claimed as there are no in vitro data of inhibition of FtsZ polymerization in presence of compound 11.

The sentence was rephrased as there are no in vitro data. However, there are strong evidences that derivative 11 is an inhibitor of the polymerization of FtsZ, a crucial step in bacterial cell division, explaining its antimicrobial activity, obtained by in silico analysis (see response 3 to Referee 3).

Specific comments for Materials and Methods

  1. In line #509-511, the authors wrote “bacterial cultures were routinely grown on 5% blood agar plate while fungal cultures on Sabouraud Dextrose Agar (Biolife Italiana S.r.l., Milan, Italy).” – I recommend adding the incubation temperature for better record keeping.

Done.

  1. Typo in line #518, the sentence should start with “The” instead of “Th”.

Done.

  1. In line #521-525, I recommend reporting the testing range of all compounds in µM instead of µg/mL for consistency.

Done.

  1. In line #544, the authors wrote “IC50 values were measured for active compounds at 100 µM.” – IC50 values are usually obtained from a dose response curve using multiple drug concentrations. I believe it should be “Microbial cell viability was measured…”

We agree with the referee, the sentence was confusing. We completely revised this point.

  1. In line #574, formatting issue. There should be one period after “…[30].”

Done.

  1. In line #577, formatting issue. “104 cells/well” should be “104 cells/well” or “10,000 cells/well”.

Done.

  1. In line #598-599, the authors wrote “…the supernatants were transferred into a clear 96-well plate, and OD405nm was read.” – please explain how does the OD405nm values of supernatants represent the lysed hRBC and please provide reference.

Done. The principle of the method was added together with the reference [18].

Specific comments for Supplementary Information

  1. In Table S1 on Page 20, the table title is unclear. What are those percentage values indicate? % of Lysed cells or % of healthy cells?

The title of the Table S1 was completely rephrased for a better understanding of readers.

Revisions are highlighted in the new version of the manuscript.

Reviewer 2 Report

Comments and Suggestions for Authors

1.       What is exactly the data in the table 1 indicated? Is it MIC data? If yes, please mention it on the title of the table.

2.      It will be great to mention statical ANOVA analysis for your experiments

3.      The positive and negative control strain is not used in the antibiofilm assay.

4.      How did you determine that the target of the compound 11 is FtsZ protein? Why did you exactly choose FtsZ protein?

Comments on the Quality of English Language

English language is good. But some sentences are quite long and not clear. It will be better to explain the methods with short and clear sentences.

Author Response

Point-by-point letter of answers to the Reviewer 2’s comments

  1. What is exactly the data in the table 1 indicated? Is it MIC data? If yes, please mention it on the title of the table.

The title of Table 1 was modified. Data are the percentage values of both microbial and Vero cell growths.

  1. It will be great to mention statical ANOVA analysis for your experiments

A paragraph describing the statistical analysis was added (paragraph 4.8.) and Figure 5 was modified by adding the result obtained by ANOVA test.

  1. The positive and negative control strain is not used in the antibiofilm assay.

      We specified in Material and Methods the controls, and we added information in Figure 5.

  1. How did you determine that the target of the compound 11 is FtsZ protein? Why did you exactly choose FtsZ protein?

Literature reports scientific evidence for the potential role of the heterocyclic compounds as FtsZ inhibitors [8-11]. In particular, ref. 11 (Lian, Z.-M.; Sun, J.; Zhu, H.-L. Design, Synthesis and Antibacterial Activity of Isatin Derivatives as FtsZ Inhibitors. J. Mol. Struct. 2016, 1117, 8–16, doi:10.1016/j.molstruc.2016.03.036) reports the antibacterial activity of isatin derivatives and suggest the mechanism of action by targeting FtsZ protein. Thus, considering that we previously identified isatin-hybrids (6k and 6m) endowed with excellent inhibitory activity against Staphylococcus aureus [ref. 15], the present paper continues our study on the rational design of isatin bis-imidazothiazole hybrids inspired to these lead compounds and adds knowledge on the possible mechanism of action of our best-performing analogue by in silico analysis.

English language is good. But some sentences are quite long and not clear. It will be better to explain the methods with short and clear sentences.

The text was revised following these suggestions.

Revisions are highlighted in the new version of the manuscript.

Reviewer 3 Report

Comments and Suggestions for Authors

The manuscript entitled “Isatin bis-imidathiazole hybrids identified as FtsZ inhibitors with on-target activity against Staphylococcus aureus” submitted by Rita Morigi et al., has been reviewed. The manuscript deals with the synthesis and in vitro activity of compound, further supported by in silico analysis. The manuscript is quite interesting and have potential results. I have few minor concerns in the presented results of the manuscript.

1.       The results of the molecular dynamics simulation should be included with relevant RMSD, RMSF, RG, SASA, intra/inter hydrogen bond contacts. Based on these information only the MD interpretation in the manuscript can be evaluated, without which evaluation of MD part in the manuscript is not possible.

2.       The free energies of binding of the compound 11 is reported as DG = -13.0 kcal mol-1 (interdomain cleft) and DG = 9.4 kcal mol-1 (GTP-binding pocket). Among which binding of 11 in interdomain cleft seems favourable. However, the DG = -13.0 kcal mol-1 is considered as less. Whether the minimal DG vales represent the weak binding? Free energy binding refers to the energy required to separate the ligand from the protein, if the value is minimal, then the ligand bound complex might be unstable or weak. How do authors consider this? Justify.

3.       Energy values usually be in negative, however represent in positive refers the weak binding. In the case of dimerization energy, authors reported binding free energy of +31.2 kcal mol-1 (positive value). Cite proper references which support the claims for this positive value and stability. Justify.

Author Response

Point-by-point letter of answers to the Reviewer 3’s comments

The manuscript entitled “Isatin bis-imidathiazole hybrids identified as FtsZ inhibitors with on-target activity against Staphylococcus aureus” submitted by Rita Morigi et al., has been reviewed. The manuscript deals with the synthesis and in vitro activity of compound, further supported by in silico analysis. The manuscript is quite interesting and have potential results. I have few minor concerns in the presented results of the manuscript.

  1. The results of the molecular dynamics simulation should be included with relevant RMSD, RMSF, RG, SASA, intra/inter hydrogen bond contacts. Based on these information only the MD interpretation in the manuscript can be evaluated, without which evaluation of MD part in the manuscript is not possible.

We followed the suggestion of the reviewer and we added in the Supplemental Material the requested analysis. The analyses are now reported in Figures S3-S7 and Tables S3-S13. The main text has been updated and now in the Material and Methods section, at paragraph 4.9, we added a sentence that reads “Post-processing analysis of the MD trajectories, such as root of mean squared deviation (RMSD), root of mean squared fluctuation (RMSF), gyration radius (RG), Solvent Accessible Surface Area (SASA) and hydrogen bon analysis were carried out employing CPPTRAJ and the results are collected in the Supplementary Material (Figures S3-S7 and Tables S3-S13)”.

  1. The free energies of binding of the compound 11 is reported as DG = -13.0 kcal mol-1(interdomain cleft) and DG = 9.4 kcal mol-1 (GTP-binding pocket). Among which binding of 11 in interdomain cleft seems favourable. However, the DG = -13.0 kcal mol-1is considered as less. Whether the minimal DG vales represent the weak binding? Free energy binding refers to the energy required to separate the ligand from the protein, if the value is minimal, then the ligand bound complex might be unstable or weak. How do authors consider this? Justify.

In this case the situation is quite simple, we are discussing about DG. When this value is negative the binding occurs (compound 11 in the interdomain cleft), when this value is positive (compound 11 in the GTP-binding pocket) the binding is thermodynamically unstable, and with longer simulation time the exit of the ligand from the binding pocket is expected. So we can state with certainty that the compound 11 prefer to bind on the interdomain cleft because in one case we have a negative value of DG and in the other case we have a positive value of DG.

In addition, between the positions, we observe an energetic difference of 22.4 kcal mol-1.

  1. Energy values usually be in negative, however represent in positive refers the weak binding. In the case of dimerization energy, authors reported binding free energy of +31.2 kcal mol-1 (positive value). Cite proper references which support the claims for this positive value and stability. Justify.

As in the previous case, negative values of DG indicate a favourable binding, while positive values of DG indicate an unfavourable binding. The interaction between two apo-FtsZs is only transient, being the DG of -3.4 kcal mol-1, a value close to zero. By adding GTP and Ca2+, that experimentally are crucial to promote the polymerization of FtsZ, the binding between two FtsZ proteins is strongly improved, reaching a value of DG = -47.9 kcal mol-1. This result confirms that both GTP and divalent cations are necessary for the polymerization of FtsZ. When inhibitor 11 binds to the interdomain cleft, the binding energy between two FtsZs monomers become positive (DG = +31.2 kcal mol-1 ), this means that the binding between the two monomer become thermodynamically unstable, and the two proteins are destined to detach themselves. In short, 11 is an inhibitor of the polymerization of FtsZ, a crucial step in bacterial cell division, explaining its antimicrobial activity.

Revisions are highlighted in the new version of the manuscript.